# Exploring the Planning and Configuration of the Hospital Wayfinding System by Space Syntax: A Case Study of Cheng Ching Hospital, Chung Kang Branch in Taiwan

**Ming-Shih Chen, Yao-Tsung Ko \* and Wen-Che Hsieh**

Department of Industrial Design, Tunghai University, No. 1727, Sec. 4, Taiwan Boulevard, Xitun District, Taichung City 40704, Taiwan; msc@thu.edu.tw (M.-S.C.); g03747007@gmail.com (W.-C.H.)

\* Correspondence: mike.ko@thu.edu.tw; Tel.: +886-4-23594646 (ext. 27)

**Abstract:** With regard to the outpatient areas of a hospital, the smoothness of the route is now taken into consideration in the process of configuring the wayfinding system. As patients often spend time on ineffective wayfinding processes, and there is limited manpower at hospitals and a lack of clarity in the information provided by the wayfinding system, it is difficult to provide effective and timely consultation services for patients. This study was conducted at Cheng Ching Hospital, Chung Kang Branch (CCH/CKB) in Taiwan. This study attempts to investigate the relationships between the wayfinding system of the outpatient areas and the patients' behaviors in the hospital. Depthmap software based on space syntax is adopted to assist in the route analysis and wayfinding behaviors. It integrates axial mapping analysis and isovist analysis and gives suggestions on the location, format and content of the wayfinding system. The final results of the study show that in the wayfinding task experiment gender has no significant impact on the effect of wayfinding efficiency, while a significant difference is found for age. Older people need more time to complete the wayfinding task, which means that they have poorer performance in wayfinding efficiency. The analysis of the results of space syntax shows that a good wayfinding system should be a symmetric tree-branch structure rather than circular structure in a medical building, that areas where it is easy to become lost should have a clear signage guiding system planning and configuration, and that clear guidance information should be provided to the patients to achieve the goal of saving consultation time and improving the quality of the medical environment.

**Keywords:** health care facilities; outpatient areas; wayfinding system; space syntax; wayfinding behaviors

## 1. Introduction

Nowadays, medical treatment is naturally more advanced, and the types of diseases and treatment methods used are becoming more and more detailed. Hence, the space required for the corresponding examination and treatment in different units has been expanded at health care facilities in order to meet the demands for medical treatment. Facing these changes in medical operations and more detailed diagnostic units, if health care facilities do not have additional land or sufficient funds to construct a new medical building, then when it comes to the early-stage wayfinding system configuration and planning the primary consideration is indoor partitioning and changing the routes in existing buildings to meet the new demands. In such cases the changed units receive additional signage to aid indoor route planning, but this often lacks overall consideration and ignores the different needs of different groups of people, resulting in great difficulties for patients when using the wayfinding system. A good wayfinding system in outpatient areas should place signage in appropriate locations to guide patients to quickly reach the required area to receive medical services. Routes and signage are both environmental factors in wayfinding processes [1]. It is thus necessary to construct a user-friendly framework for an indoor

public space wayfinding system for health care facilities, for use by both patients and visitors [2,3].

Blumberg and Devlin [4] pointed out that the amount of wayfinding behaviors among hospital users is an important indicator of the pressure put on hospital consultations. In a complex public environment, all such behaviors must be guided by information and intelligence objects in the environment to be successfully completed. Chen [5] noted that one way to handle wayfinding problems is related to the use of directories. Baskaya et al. [6] indicated that space syntax and wayfinding systems are important environmental factors influencing indoor wayfinding. To reduce the occurrence of wayfinding problems, it is necessary to better understand the characteristics of wayfinding behaviors, the space syntax in which these occur, and to help people find their destinations through appropriate signage formats, thus addressing the problems people have with wayfinding and providing users with good spatial service quality. Huang and Tseng [7] stated that the research on wayfinding signage design in medical environments in Taiwan is relatively limited, with most such studies examining the addition or expansion of wayfinding systems. They also stated that in the future more emphasis should be given to research on insufficient directional information and the decision points in the wayfinding behaviors.

Based on above, we attempted to investigate the content and form of the signage configuration in this research context, and further applied the space syntax theory and Depthmap software to assist in the simulation analysis of the routes in order to find out the wayfinding problems caused by the current spatial route planning in health care facilities. Based on the results we proposed some suggestions for wayfinding system planning and configuration for hospitals, thus achieving the purpose of this study.

## 2. Related Works

### 2.1. Characteristics of the Spatial Environment

Some researchers indicated that even well-designed signs often do not provide enough information to guide visitors to move freely in a hospital without becoming lost [8,9]. Carroll [10] noted that floor plan complexity has a significant effect on wayfinding behaviors, while O'Neill [11] suggested a close relationship between building plan complexity and the number of times people become lost. The sizes, forms and relationships among spaces and the irregularity of floor plans are the main causes of this complexity. Moderate and straightforward route planning in space will help reduce the problems of wayfinding, although too many route intersections will increase the number of decision points, making users more easily confused. Best [12] pointed out that too many decision points increase the uncertainty of users' judgments, creating confusion and increasing the difficulty of finding the way. Thus, too many decision points on a floor plan will cause difficulties in wayfinding. In fact, any point in the wayfinding process can be a decision point for behavior, which is not limited to the choice of directions. The lack of complete information in the environment also prevents users from constructing a complete spatial concept, resulting in an inability to recognize and analyze the situation when searching for directions [13]. Arthur and Passini [14] proposed decision making as a means for solving wayfinding spatial problems and determining decision execution and information processing, including environmental awareness and cognition. When planning and designing a hospital wayfinding system, it is necessary to consider both patients' perceptions and the impact on the hospital space environment. For patients, good design can promote recovery, because when patients understand their environment and try to find their destination in a more relaxed way, it can re-establish a sense of control and accomplishment for them [15,16].

### 2.2. Space Syntax Literatures

Space syntax theory is a method of "quantifying the environment as a set of predictive variables for specific behaviors" that predicts wayfinding behaviors in public areas by targeting people's tendency to move to spaces with higher levels of integration; in other words, such spaces act as transit hubs and are more connected to other places [17]. This

principle has been demonstrated in various architectural layouts. Morgareidg et al. [18] also applied space syntax to analyze emergency rooms for environmental planning and conducted research on the interaction situation with regard to users' actual routes. Wang [19] used space syntax to investigate the factors impacting wayfinding that resulted from the layout inside a museum. The results of axial mapping analysis and space syntax indicate that the longest routes have a higher proportion of wayfinding behaviors in the case of a moderate Rn value of route configuration characteristics, but the interlacing of complex routes and excessive number of decision points in a floor plan increase the chance of becoming lost significantly when the number of nodes (CN) of the moderate Rn value is six or more. If the environment does not provide complete information, the pathfinder cannot construct a complete spatial concept or carry out cognitive analysis, and thus cannot make correct wayfinding decisions. Chaudhary et al. [20] applied the concept of space syntax to describe the configuration of medical buildings, such as making predictions on the movement of nurses through hospital units, by measuring their connectivity and integration and analyzing spatial configurations (i.e., layouts). Haq and Luo [21] pointed out that many studies have used this technique to examine the spatial configurations of health care facilities, including wayfinding systems.

### 2.3. Wayfinding Behavior

The term "wayfinding" can be traced back to the 1960s when it was first mentioned by Kevin Lynch [22] in his book *The Image of the City*. Lynch [22] used "wayfinding" to explain people's cognition and analytical capabilities toward the urban environment. The elements that prevent people from becoming lost in the urban space were classified through a process of experimentation, including: (1) orientation recognition; (2) environmental perception; (3) strategy adoption; (4) gender and age characteristics. Early environmental psychologists focused on exploring the relationships between "space" and "behavior", trying to clarify the decision-making processes about how people will react and find the right route when they become lost in space and people's perception, cognition, and route choice. Downs and Stea [23] stated that wayfinding behavior is a process of people understanding which external environment they are in using a certain method and making route choices based on this. Evans et al. [24] defined wayfinding behavior as "a work of recognizing and reacting to the complex space environment and guiding sign systems" from the perspective of human perception. Blades [25] explained people's wayfinding behavior in the environment based on them learning the characteristics of the current surrounding environment and recalling past experiences, and constructing and choosing the correct routes from the starting point to the destination. Some studies have claimed that men's wayfinding ability is better than women's, due to the differences in preference between men and women in the wayfinding strategies adopted. Men usually use an Euclidean strategy while women generally use a landmark one to find their ways. However, if the focus is on the time spent on the wayfinding process and the degree of hesitation and delay at decision points, there is no significant difference between men and women [26,27]. As for the relationship between age and wayfinding ability, most studies believe that older people have worse performance with regard to wayfinding [28,29]. In general, wayfinding behavior is not based on a single, linear information processing action, but is a recursive information process. The decision-making processes interact with each other at the psychological level after a person receives various stimuli based on environmental information, including the recognition of the complex structure of the spatial environment and the graphics of guiding sign systems.

### 2.4. Decision-Making and Wayfinding Systems

MacMinner [30] suggested that people use architectural space and internal information to determine their own location, indicating that "architectural space" and "internal processing" can be used as vision clues for users to identify their orientation. In architectural spaces, such as staircases, elevators, halls and corridors, spatial characteristics are

used to identify orientation. Using the clues provided by the environment can reduce the chances of becoming lost. As for internal processing, this includes wall color, form and structure, floor changes, and the use of lighting techniques to emphasize or hide certain areas, such as ceiling treatments and the configuration of equipment. Rousek and Hallbeck [31] pointed out that a well-designed guiding wayfinding system for health care facilities should take into account the internal and external environment, pay attention to relevant design elements to improve understanding, and integrate visual guidelines with the design concept to reduce the pressure and anxiety of the people attending a clinic and, thus, achieve the greatest independence of the individual. Koneczny et al. [32] stated that improper design planning will cause wayfinding-related difficulties for the public, such as: (1) improper decorative elements such as luminous floor tiles; (2) too bright or too dark lighting may be misleading; and (3) too small a signage size and improper placement, along with many other factors.

In the past, most studies on wayfinding behaviors and signage systems used questionnaires combined with on-site examinations of the actual behaviors of experimental participants. However, traditional experiments and questionnaires are time-consuming and labor-intensive, and often result in human interference that impacts the findings of the research. Nowadays, with the advance of digital technology, computer techniques can provide researchers with more accurate and efficient assistance in the study of wayfinding behaviors. Moreover, the use of data to analyze the deeper meaning of spatial complexes and to carry out image analysis can help explain people's movement habits and navigation needs in space. Therefore, in this study, we aim to use the Depthmap analysis software with space syntax as a tool for analysis and prediction to find the points where people tend to become lost in order to provide useful references for health care facilities in designing and planning their wayfinding systems.

## 3. Methodology

In this study, we adopted the axial mapping analysis and isovist analysis of Depthmap software based on space syntax to define the routes people use in health care facilities and predict the frequency and visual location of each route. The results of the axial mapping and isovist analysis are applied to provide references for the relative health care facilities to establish their wayfinding systems.

### 3.1. Research Field

In this study, Cheng Ching Hospital, Chung Kang Branch (CCH/CKB), Taichung, Taiwan was taken as the research field, with the focus on the outpatient medical space from the first basement floor to the second floor of the Chung Kang Building. The spatial configuration and composition characteristics of the research field are described below:

(1)　First floor of Chung Kang Building

The first floor of Cheng Ching Hospital, Chung Kang Branch is the entrance/exit for the patients. The entrance of main route is located at Fukang Rd. The major structure is centered on a high-ceilinged lobby. The lobby itself also serves as the mobile space of the institute and the medical collection and waiting area. Therefore, after entering the lobby you have the pharmacy (medicine collection office) on your left and the united service center on the right, which is used for first-time patients to find the consultation unit, make an appointment and give basic information.

(2)　Second floor of Chung Kang Building

This floor is divided into mental health, dentistry, otorhinolaryngology, psychiatry, and infectious disease units with a semi-open waiting area connecting to private treatment spaces.

(3)　First basement floor of Chung Kang Building

The main consultation units and examination rooms as well as the appointment and cashier counter are located on the first basement floor. The outpatient areas include internal

medicine, medical, surgery, family medicine, rehabilitation, and health care consultation rooms, while the examination rooms include general physiological examination rooms, blood sampling stations, cardiology-related examination rooms, extracorporeal shock wave rooms, urodynamic rooms, and radiology-related examination spaces.

### 3.2. Space Syntax Theory

Shu (1999) analyzed the deeper meaning of space syntax. The foundation of its inner structural logic can be divided into two directions: relative depth and choice of path. Relative depth refers to the relative position and connection relationships among the elements of the structure in the space syntax system, and from this is derived a depth relationship expression where each route can be regarded as a unit in the dynamic space syntax (Figure 1), and this is used to discuss the depth relationship, while the path choice is based on all possible paths for the intercommunication between any two elements in the space syntax system. Although there may be more than one path choice between two elements in the system, the relative depth relationship between them is unique, i.e., the shortest path depth between them. Taking the example in this study, there are many paths from the lobby of the Chung Kang Building to the appointment and cashier counter on the first basement floor for path choice; however, from the analysis of relative depth, the relative depth is the layer of the shortest routes to reach the destination [33].

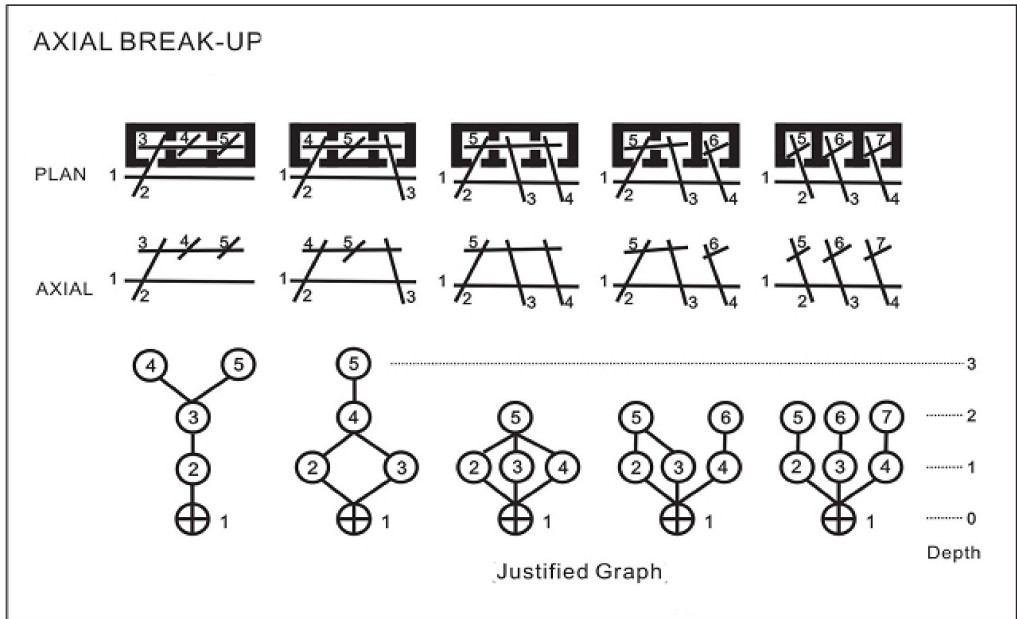

**Figure 1.** Dynamic space syntax graph [34].

The element, "longest route", comes from two perspectives, visibility and permeability, both of which are related to innate human instincts. In this study, the concept of the decomposed elements of the longest route was used to analyze the outpatient area routes of the health care facilities. As shown in Figure 2, in the spatial floor plan configuration, by plotting the longest route connecting units we can develop an axial map. After analyzing this, we can obtain the integration of each longest route presented by color code, ranging from red to blue (high to low integration). Hence, we can learn from an axial map that the longest route 1 (red lines) is the most convenient location in the area, while the longest route 12 (blue lines) is in the least convenient location.

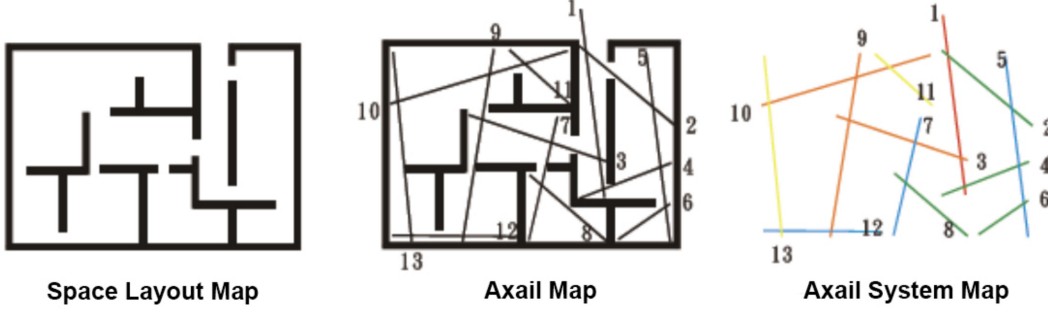

**Figure 2.** Analysis of the axial map [34].

A detailed introduction to the related proper nouns of space syntax theory is given below [35]:

Rn: Rn is the meaning of "global integration" in space syntax. Rn is used to calculate the deep or shallow relationships between each space or each axis. The spaces or axes with high global integration are those that reach all other spaces or axes with relatively few steps. That is to say, these spaces or axes with high global integration are distributed in convenient locations that are easier to reach in the spatial organization.

CN: CN means the connectivity between spaces and axes. It is used to calculate the sum of each space or axis connected to other spaces or axes.

Axial map/Axial mapping analysis: Space syntax theory compares the axis of space to the movement of a person in space. Each axis represents a number of steps in space, and movement also represents the visibility of a person in motion. In principle, a fewer number of axes is better, and a longer length of axes is better. Axial mapping analysis is used to present the deduction process of the overall spatial axis map in the most concise way.

Isovist analysis: Isovist analysis is also called visual graph analysis. Any position in the space can produce its isovist. Its definition is the maximum visual range we can see from a certain position. Isovist analysis is based on the integration to calculate the spatial neighborhood size and clustering coefficient and the mean shortest path length boundary. In this way, a visual image of the space is generated. Therefore, isovist analysis can show the relationships between visibility and permeability in space. The characteristics of visual graphics are closely related to the manifestations of space perception, such as wayfinding, movement and space use.

## 4. Experiments and Analysis Results

In this study, the CCH/CKB medical building was taken as the major research field. In the physical investigation, the outpatient areas from the first basement floor to the second floor of the medical building were used as the experiment site. In the first part of the study, we investigated the distribution and location of consultation rooms and examination rooms in the outpatient areas and recorded the type of signage and locations, then we applied space syntax to draw the paths and routes to anticipate the user behavior tendency and visual concentration areas. The experimental results of the participants' wayfinding tasks were compared to the analysis results predicted by space syntax, to see if they are consistent. Finally, we applied axial mapping analysis and isovist analysis to make a consolidated summary in order to give suggestions on planning the wayfinding system at the research field.

### 4.1. Outpatient Areas Space Configuration and Signage Forms

In this paper, we first analyzed and annotated the signage configuration and forms from the first basement floor to the second floor of the outpatient areas of the building, and used different symbols to represent the different attributes of the signage, and finally summarized them into two major blocks: (1) the contents of the signage: identification,

guidance, direction and description, in total four types (Figure 3); (2) the forms of the signage: free stand, up hanging, protruding and wall-mounted, in total four types (Figure 4).

|  | (A) Identification | (B) Guidance | (C) Direction | (D) Description |
|---|---|---|---|---|
| Free stand | A | B | C | D |
| Wall-mounted | A | B | C | D |
| Up hanging | A | B | C | D |
| Protruding | A | B | C | D |

**Figure 3.** Classification of the content of the signage in the research field.

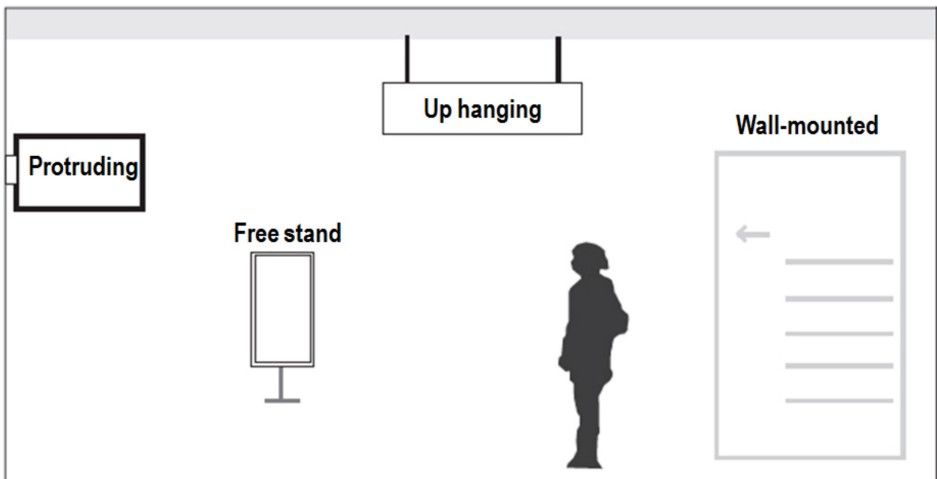

**Figure 4.** Classification of the forms of the signage in the research field.

We actually went to CCH/CKB hospital to do fieldwork. Based on the above classification of the content and presentation format of signage, we investigated and recorded the signage configuration and distribution in the outpatient area from the first basement floor to the second floor. The detailed information is marked on the layout diagram of each floor. The analysis results are shown in Figure 5.

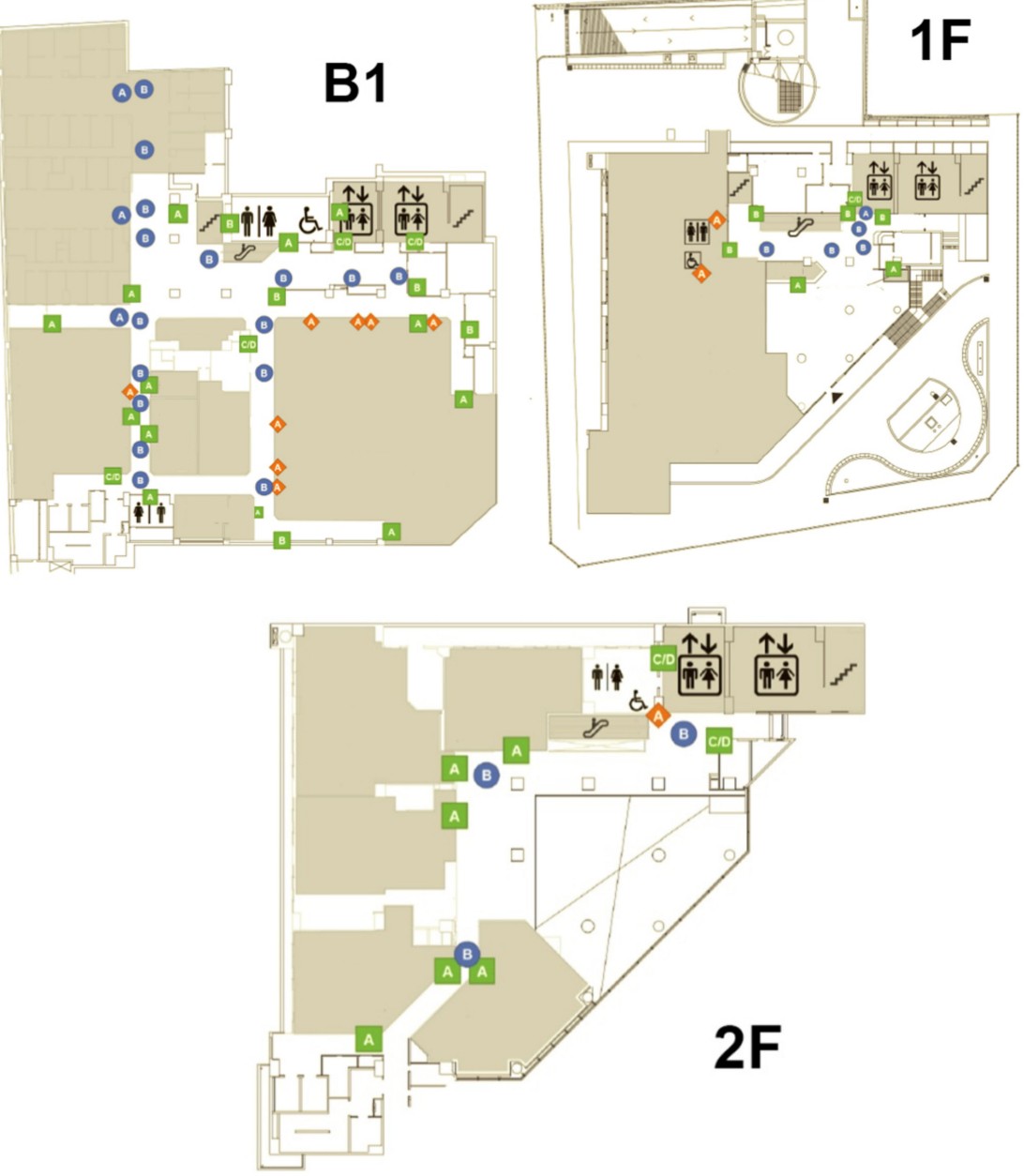

**Figure 5.** The signage configuration and distribution of the outpatient areas B1 to 2F in CCH/CKB.

The results showed that there are seven categories and 82 signs in the outpatient area from the first basement floor to the second floor. Among them, the distribution of the up hanging type with guidance contents has a maximum of 23 signs, accounting for 28.05% of the total. Next is the signage of the wall-mounted type with identification contents with 21 examples, accounting for 25.61% of the total. In addition, in terms of floor distribution, the number and types of the signage are the most on the first basement floor due to its larger area and more clinic and examination rooms, with a total of seven categories and 53 signs. The number and type distribution of the signage on each floor is shown in Table 1.

**Table 1.** Statistics on the number and type distribution of the signage on each floor.

| | Wall-Mounted | | | | Up Hanging | | Protruding | |
|---|---|---|---|---|---|---|---|---|
| | **(A)** Identification | **(B)** Guidance | **(C)** Direction | **(D)** Description | **(A)** Identification | **(B)** Guidance | **(A)** Identification | **Total** |
| | A | B | C | D | A | B | A | |
| B1 | 13 | 5 | 4 | 4 | 3 | 16 | 8 | 53 |
| 1F | 2 | 4 | 1 | 1 | 1 | 4 | 2 | 15 |
| 2F | 6 | 0 | 2 | 2 | 0 | 3 | 1 | 14 |
| Sum | 21 | 9 | 7 | 7 | 4 | 23 | 11 | 82 |
| Percentage | 25.61% | 10.97% | 8.54% | 8.54% | 4.88% | 28.05% | 13.41% | |

### 4.2. The Results of Axial Mapping Analysis

#### 4.2.1. The First Floor of CCH/CKB Medical Building

The main entrance/exit route of CCH/CKB Building is the first floor, and the main building is accessed via route number 62. For direct access to the outpatient areas on the second floor, the vertical routes available are escalator route 60, elevators 235–237 and 232–234; staircases are also available, with route 246 to the second floor. The most common routes on the first floor of the building can be seen in Figure 6, and the properties of each route can be compared in Table 2. The routes in each section are numbered by Depthmap software, and the global integration is ranked by the Rn value of all the outpatient area routes in the medical building after axial mapping analysis. A larger Rn value indicates higher integration and a rank closer to the top; in contrast, a smaller Rn value with lower integration is ranked towards the bottom. From the axial mapping analysis of the first floor of the CCH/CKB Building in Figure 6, we can predict that the escalators to the first basement floor of the medical building with the route numbers 144 and 221 are ranked 12 and 15 in terms of global integration, i.e., in the top 15 most frequently used routes for direct access (Table 2).

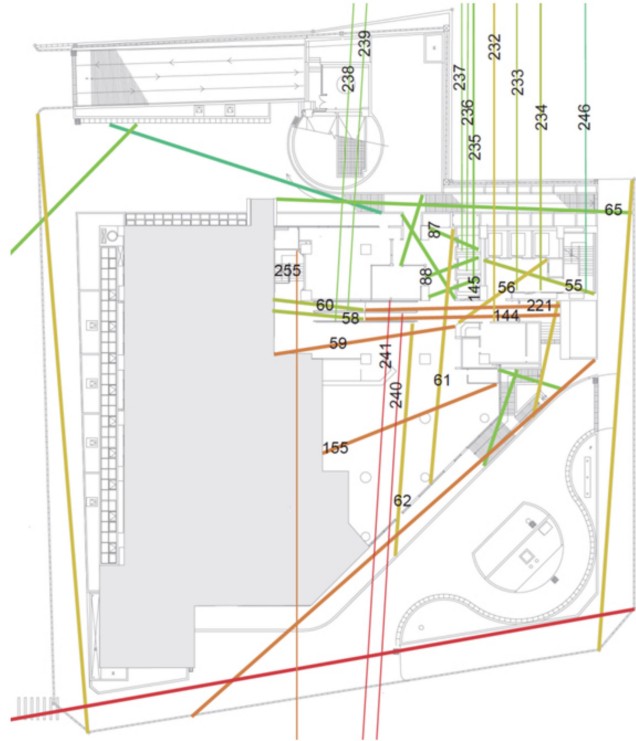

**Figure 6.** The axial mapping analysis of the first floor of the CCH/CKB Building.

**Table 2.** The quantification of the dynamic syntax of the major and vertical routes on the first floor of the CCH/CKB Building.

| Rank | Line | Route Properties | Rn | CN |
|------|------|------------------|-----|-----|
| 1 | 59 | Passage to 2F Escalator | 1.1866 | 5 |
| 2 | 255 | Staircase a | 1.1432 | 7 |
| 3 | 144 | Escalator 1F to B1 | 1.1209 | 4 |
| 4 | 155 | Lobby routes | 1.1190 | 4 |
| 5 | 221 | Escalator B1 to 1F | 1.1003 | 3 |
| 6 | 62 | Main entrance/exit routes | 1.0645 | 3 |
| 7 | 61 | Lobby route-elevator area | 1.0588 | 8 |
| 8 | 60 | Escalator 1F to 2F | 1.0030 | 2 |
| 8 | 58 | Escalator 2F to 1F | 1.0030 | 2 |
| 10 | 232 | Elevatorhall B-Vertical route_Elevator d | 0.9659 | 8 |
| 11 | 87 | Elevator hall A—Elevator c | 0.9428 | 2 |
| 12 | 88 | Elevator hall A—Elevator b | 0.9213 | 3 |
| 13 | 145 | Elevator hall A—Elevator a | 0.9207 | 3 |
| 14 | 237 | Vertical Route_Elevator c | 0.9195 | 4 |
| 15 | 235 | Vertical Route_Elevator a | 0.8967 | 3 |
| 15 | 236 | Vertical Route_Elevator b | 0.8967 | 3 |
| 17 | 233 | Elevatorhall B-Vertical route_Elevator e | 0.8507 | 6 |
| 18 | 234 | Elevator hall B-Vertical route_Elevator f | 0.8507 | 6 |
| 19 | 55 | Elevator hall B-Staircase b | 0.8242 | 5 |
| 20 | 56 | Aisle-Elevator hall B | 0.8237 | 4 |
| 21 | 246 | Staircase b | 0.7444 | 3 |

(Rn: global integration; CN: connectivity numbers between spaces and axes).

### 4.2.2. The First Basement Floor of the CCH/CKB Medical Building

The main consultation units and examination rooms as well as the appointment and cashier counter are located on the first basement floor of the Chung Kang Building. The outpatient area includes internal medicine, medical, surgery, family medicine, rehabilitation, and health care consultation rooms, while the examination rooms include general physiological examination rooms, blood sampling stations, cardiology-related examination rooms, extracorporeal shock wave rooms, urodynamic rooms, and radiology-related examination spaces. The general public can reach this floor by vertical routes with escalator routes 34 and 33, elevators 235–237 and 232–234, or by stairs with route 246. From Figure 7, we can find that the main moving routes are numbered 32, 193, 186, and 8, and their Rn rank is 3, 6, 7, and 5, respectively (Table 3), which can be used to predict that this floor is the main queuing area for people attending the clinic. In the structure formed by the main moving routes, it is found that the route numbers 32, 186, 8 and 2 form a symmetrical ring structure.

### 4.2.3. The 2nd Floor of CCH/CKB Medical Building

On this floor, the units are infectious disease, neurology, otorhinolaryngology, mental health and dentistry with a semi-open waiting area connecting to private treatment spaces setting the boundary between the different units (Figure 8). The open type of routes are numbered 100, 113 and 50 (Table 4) with a single moving route, which can be regarded as the trunk routes, while the semi-open type of routes of the clinic with the route numbers 100–111, 100–107 and 100–113, and routes 50–86 and 50–52, are determined as the branch routes based on their route properties. Under the overall review, the route plan of the 2nd floor is a single type of tree structure, a simple and concise space pattern, which is a more user-friendly spatial route plan for the patients.

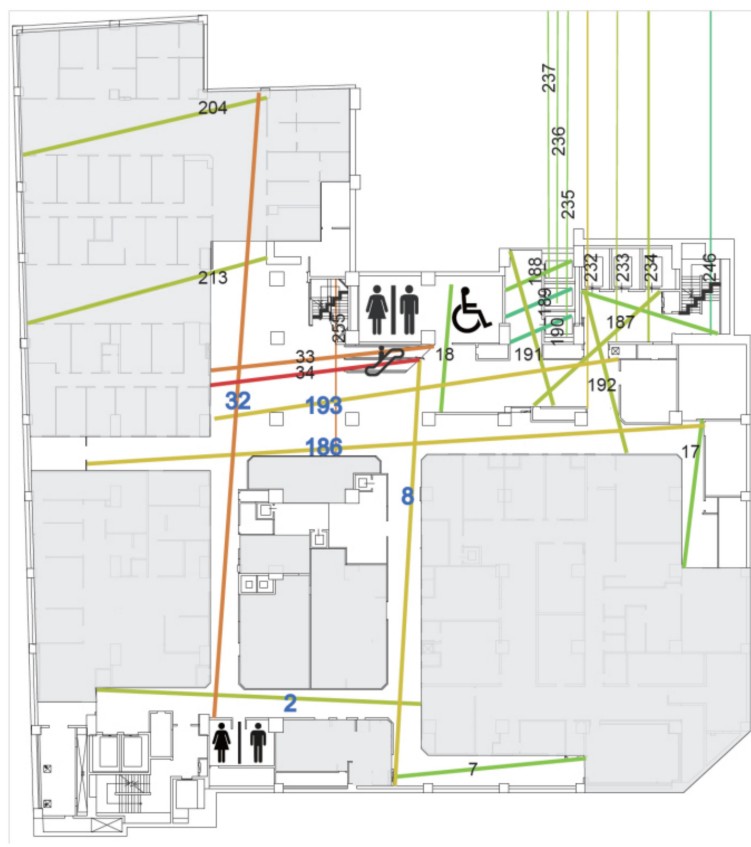

**Figure 7.** The axial mapping analysis of the first basement floor of the CCH/CKB Building.

**Table 3.** The quantification of the dynamic syntax of major routes on the first basement floor of the CCH/CKB Building.

| Rank | Line | Route Properties | Rn | CN |
|------|------|------------------|------|------|
| 1 | 34 | Escalator 1F to B1 | 1.2181 | 4 |
| 2 | 33 | Escalator B1 to 1F | 1.2021 | 4 |
| 3 | 32 | Outpatient areas—Physiological examination room (Diagnostics 30) | 1.1557 | 15 |
| 4 | 255 | Staircase a | 1.1432 | 5 |
| 5 | 8 | Radiology counter—Aisle of urodynamic examination room | 1.1038 | 13 |
| 6 | 193 | First basement floor corridor-elevator area | 1.0745 | 8 |
| 7 | 186 | Machine for making an appointment, cashier and taking a number—64-row computed tomography (CT) room | 1.0427 | 17 |
| 8 | 2 | Front passage of extracorporeal shock wave lithotripsy room | 0.9693 | 10 |
| 9 | 232 | Elevator hall B-Vertical route_Elevator d | 0.9659 | 8 |
| 10 | 191 | Passage—Elevator hall A | 0.9612 | 5 |
| 11 | 187 | Aisle—Elevator hall B | 0.9460 | 7 |
| 11 | 192 | 64-row computed tomography (CT) room—Elevator hall B | 0.9460 | 6 |
| 13 | 237 | Vertical route_Elevator c | 0.9195 | 4 |
| 14 | 236 | Vertical route_Elevator b | 0.8973 | 3 |
| 15 | 235 | Vertical route_Elevator a | 0.8967 | 3 |
| 16 | 188 | Elevator hall A- Elevator c | 0.8679 | 2 |
| 17 | 189 | Elevator hall A- Elevator b | 0.8507 | 2 |
| 17 | 233 | Elevator hall B-Vertical route_Elevator e | 0.8507 | 6 |
| 17 | 234 | Elevator hall B-Vertical route_Elevator f | 0.8507 | 6 |
| 20 | 190 | Elevator hall A—Elevator a | 0.8502 | 2 |
| 21 | 57 | Elevator hall B—Staircase b | 0.8372 | 6 |
| 22 | 246 | Staircase b | 0.7444 | 3 |

(Rn: global integration; CN: connectivity numbers between spaces and axes).

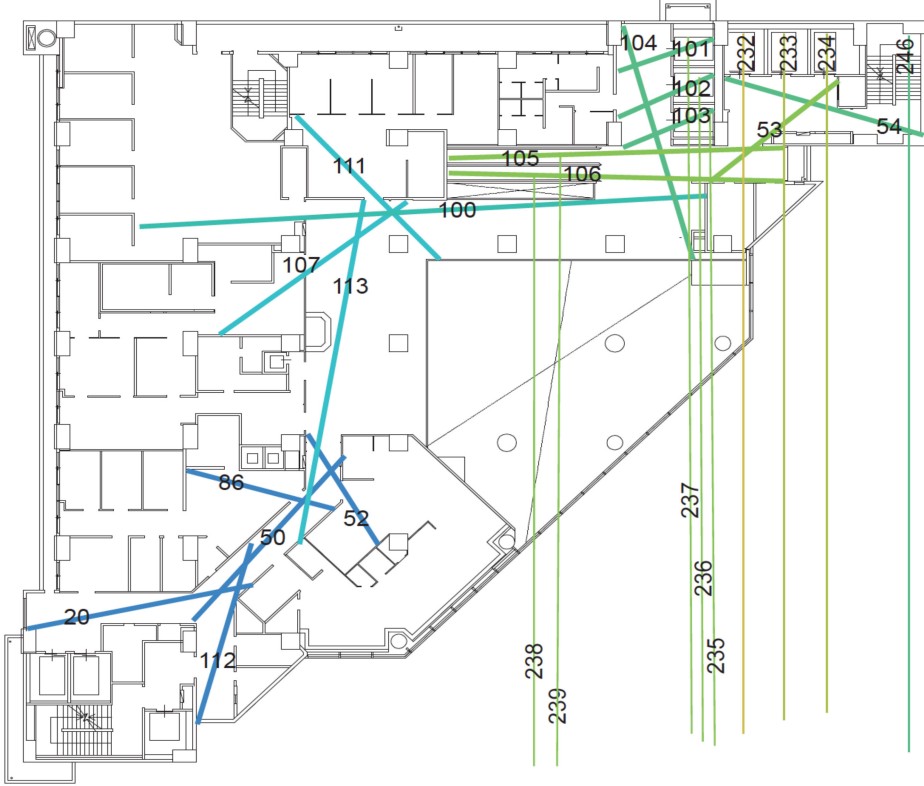

**Figure 8.** The axial mapping analysis of the second floor of the CCH/CKB Building.

**Table 4.** The quantification of the dynamic syntax of major routes on the second floor of the CCH/CKB Building.

| Rank | Line | Route Properties | Rn | CN |
|------|------|------------------|-----|-----|
| 1 | 232 | Elevator hall B-Vertical route_Elevator d | 0.9659 | 8 |
| 2 | 237 | Vertical route_Elevator c | 0.9195 | 4 |
| 3 | 238 | Vertical route_Escalator 2F to 1F | 0.9008 | 2 |
| 3 | 239 | Vertical route_Escalator 1F to 2F | 0.9008 | 2 |
| 5 | 236 | Vertical route_Elevator b | 0.8973 | 3 |
| 6 | 235 | Vertical route_Elevator a | 0.8967 | 3 |
| 7 | 53 | Aisle—Elevator hall B | 0.8717 | 6 |
| 8 | 105 | Escalator 1F to 2F | 0.8597 | 3 |
| 8 | 106 | Escalator 2F to 1F | 0.8597 | 3 |
| 10 | 233 | Elevator hall B-Vertical route_Elevator e | 0.8507 | 6 |
| 10 | 234 | Elevator hall B-Vertical route_Elevator f | 0.8507 | 6 |
| 12 | 104 | Passage—Elevator hall A | 0.8486 | 8 |
| 13 | 54 | Elevator hall B—Staircase b | 0.8460 | 5 |
| 14 | 102 | Elevator hall A—Elevator b | 0.8164 | 3 |
| 15 | 103 | Elevator hall A—Elevator a | 0.8159 | 2 |
| 16 | 101 | Elevator hall A—Elevator c | 0.8111 | 2 |
| 17 | 100 | Aisle—Neurology | 0.7596 | 8 |
| 18 | 246 | Staircase b | 0.7444 | 3 |
| 19 | 113 | Front aisle of the appointment and cashier counter | 0.6770 | 5 |
| 20 | 107 | Route 113—Otorhinolaryngology clinic | 0.6764 | 5 |
| 21 | 111 | Route 100—Infectious diseases clinic | 0.6711 | 4 |
| 22 | 50 | Common corridor of mental health and dentistry | 0.6058 | 5 |

(Rn: global integration; CN: connectivity numbers between spaces and axes).

From the analysis above, it can be found that the five routes (Table 3) which are numbered 32, 193, 186, 2, and 8 are the open main routes. From the results of axial mapping analysis in space syntax, the global integration value, Rn, is a moderate value. Moreover, the number of branches connected to each of the five open main routes exceeds six, which verifies Wang's study [19]. In his work it is pointed out that the longest routes have a higher rate of wayfinding behaviors under a moderate Rn value of route configuration characteristics. Due to the complexity of the routes and the large number of decision points on the floor plan, the chance of becoming lost increases significantly when the number of connected routes (CN value) with a medium Rn value is six or more.

*4.3. Wayfinding Tasks Experiment Planning and Analysis*

In the outpatient area of the medical building, the clinics and examination rooms are divided into different departments on different floors according to the business considerations of different departments. Meanwhile, each department has an independent diagnosis and treatment area. This can be regarded as the outpatient clinic area of each department as a spatial unit, which is then arranged and combined into the plane space of each floor. In order to objectively analyze wayfinding behaviors of the general public in the hospital, the recruitment object of this study was members of the general public who had never been to the experimental site before. The age range was between 19 and 64. The experiment participants were divided into three age groups (19–30, 31–44, 45–64). They were all assigned the same wayfinding tasks on the first floor, second floor and first basement floor of the outpatient area of the hospital. The wayfinding tasks were executed from the entrance of the first-floor gate of the hospital as a starting point. The participants had to complete the wayfinding tasks one by one in the order listed in Table 5. During the experiment, the participants could only reach the destination according to the signage guiding system of the medical building. The main task of the research recorder was to confirm whether the participants did indeed complete each wayfinding task and record the time to complete the task (Figure 9). The research recorder included two members, one of whom used the digital camera to record video images and record the whole wayfinding task of each participant, while the other followed the participants at a distance of about five meters, and observed and recorded whether the participants actually completed the wayfinding tasks for each floor. The recorder also wrote notes to record whether the participants had any wayfinding difficulties or became lost, such as whether there was a place or corner where they moved back and forth or stayed for a long time, and if so, the recorder wrote down the location and duration of stay. Afterwards, these notes were compared and analyzed with the video images. The criterion of this study is that if the experimental participant moved back and forth more than twice or stayed for more than 1 min in the same place or corner, they are judged as having lost their way. Research recorders were required to receive two hours of training and task instructions before actually performing the recording tasks to ensure that they could do their work. Finally, the time spent by the 30 participants (15 males and 15 females) when performing the wayfinding tasks was calculated after all experiments were completed. A comparative analysis of different genders and the time spent on wayfinding tasks with three different age groups (five males and females in each age group) of 19–30, 31–44, and 45–64 was also performed.

Basically, in Table 6 it can be seen with regard to the average time spent on wayfinding tasks that women need less time (mean = 1621, SD = 327) than men (mean = 1697, SD = 236). To further examine whether gender has significant differences in the time spent on wayfinding tasks the independent samples *t*-test was applied. The results also show that the significance value of 0.470 far exceeds the statistical significance value of 0.05 ($p < 0.05$). Therefore, it can be determined that the influence of gender on the wayfinding tasks is a homogeneity factor. This means that the time spent by the participants in this experiment to perform wayfinding tasks was not significantly different with regard to gender.

**Table 5.** Wayfinding task contents of the clinic and examination rooms in the different floors of the medical building.

| Floor | CCH/CKB Building | Degree of Difficulty of the Wayfinding Tasks | |
| --- | --- | --- | --- |
| | | **Mean** | **SD** |
| 1F | Information Station | 2.63 | 0.23 |
| | Pharmacy | 1.57 | 0.15 |
| 2F | Registration & Cashier | 1.85 | 0.12 |
| | Ultrasonic Room | 2.63 | 0.33 |
| | Otorhinolaryngology (Clinic 52) | 2.13 | 0.26 |
| | Dentistry | 2.15 | 0.41 |
| | Mental Health Clinic | 2.37 | 0.28 |
| B1 | Machine for making an appointment, cashier and taking a number | 2.13 | 0.16 |
| | Clinic Room 9 | 2.37 | 0.08 |
| | Clinic Room 52 | 2.05 | 0.32 |
| | Clinic Room 26 | 2.13 | 0.42 |
| | Radiology Counter | 2.37 | 0.06 |
| | Physiological Examination Room (Clinic 30) | 3.92 | 0.26 |
| | Blood Collection Station | 3.83 | 0.35 |
| | Aisle of Urodynamic Examination | 3.87 | 0.22 |
| | Extracorporeal Shock Wave Lithotripsy Room | 3.60 | 0.36 |
| | MRI Room | 3.43 | 0.15 |
| | 64-row Computed Tomography (CT) Room | 3.83 | 0.26 |
| | Angiography Room | 3.67 | 0.12 |

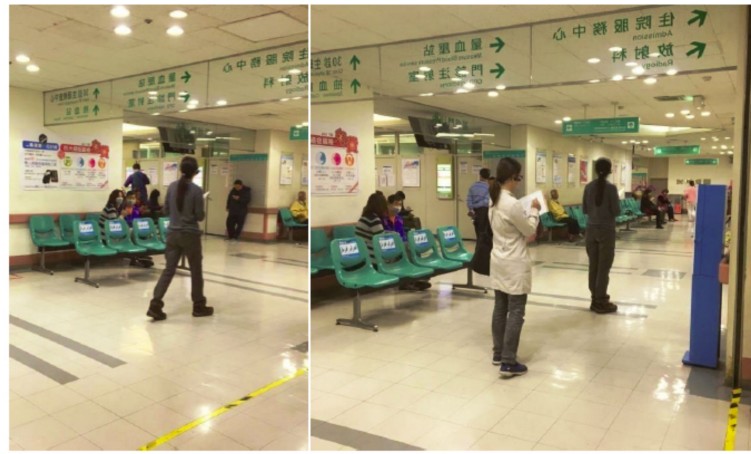

**Figure 9.** The research recorder observed and recorded the participants performing the wayfinding tasks.

In addition, it can be seen from Table 7 that with regard to the three different age groups (19–30, 31–44, and 45–64) it was the group aged 19–30 years old that had the shortest wayfinding time (mean = 1531, SD = 258), and thus was most efficient, followed by the group aged 31–44 years old (mean = 1687, SD = 231), and finally that aged 45–64 years old (mean = 1758, SD = 282). The upper and lower values of the 95% confidence interval of mean indicate that the sample follows a normal distribution. Meanwhile, the homogeneity of variances is also calculated using a Levene test, with the results shown in Table 8. The significance value of 0.815 is greater than the threshold value of 0.05. This means that these three sets of parameters are homogeneous. In order to verify whether the three different age groups have significant differences in the time spent on the wayfinding tasks, this study then used statistical one-way analysis of variance (ANOVA) to analyze the data. From the

results in Table 9 it can be seen that the significance value of *p* is 0.028 under the statistically determined value of 0.05. Further post-hoc analysis was used to determine whether there were significant differences between the groups. The results in Table 10 show that the main significant differences in the time spent on wayfinding tasks were between the groups aged 19–30 and 45–64. This means that the time spent by the participants in the experiment to perform the wayfinding task was significantly different due to the difference in age. In short, older people needed much more time to complete the wayfinding tasks, which means that they had poorer performance with regard to wayfinding efficiency. There are three possible reasons why the older participants needed to spend more time on the wayfinding tasks. The first may be the deterioration of physiological functions, especially in mobility, which is not as efficient as in younger people. The second reason may be the inability to see the signage system clearly due to poorer vision. The third is that the older participants may have had problems with reduced cognitive ability and thus, understanding of the graphic images, and so an inability to obtain the necessary and sufficient wayfinding information from the guiding signage system in order to reach the destinations in a short time. The detailed reasons for these results can be studied in future research.

**Table 6.** Statistics and *t*-test results of the time spent on the wayfinding tasks by different genders.

|  |  |  |  |  | Independent Samples *t*-Test | |
| --- | --- | --- | --- | --- | --- | --- |
|  | **Gender** | **Sample** | **Mean** | **SD** | **t** | **Sig. (p)** |
| Time spent on wayfinding tasks | Male | 15 | 1697 | 236 | 0.733 | 0.470 |
|  | Female | 15 | 1621 | 327 |  |  |

Compared significance under the 0.05 level is indicated by *. * (Unit: Second).

**Table 7.** Descriptive statistics of the time spent on wayfinding tasks for three different age groups.

| Age Range | Sample | Mean | SD | 95% Confidence Interval | | Minimum | Maximum |
| --- | --- | --- | --- | --- | --- | --- | --- |
|  |  |  |  | **Lower Bound** | **Upper Bound** |  |  |
| 19–30 | 10 | 1531 | 258 | 1251 | 1886 | 1099 | 2039 |
| 31–44 | 10 | 1687 | 231 | 1374 | 1978 | 1159 | 2165 |
| 45–64 | 10 | 1758 | 282 | 1582 | 2062 | 1470 | 2213 |
| Sum | 30 | 1659 | 257 | 1402 | 1975 | 1243 | 2139 |

(Unit: Second).

**Table 8.** Homogeneity test of variance of the time spent on wayfinding tasks for three different age groups.

| Levene Statistics | Freedom Degree of Numerator | Freedom Degree of Denominator | Sig. (p) |
| --- | --- | --- | --- |
| 0.206 | 2 | 27 | 0.815 |

**Table 9.** One-way ANOVA of the time spent on wayfinding tasks for three different age groups.

| Items | Sum of Square | Freedom Degree | Mean Sum of Squares | F | Sig. (p) |
| --- | --- | --- | --- | --- | --- |
| Between Groups | 269,885.400 | 2 | 134,942.700 | 1.771 | 0.028 * |
| Within Groups | 2,057,681.400 | 27 | 76,210.422 |  |  |
| Sum | 2,327,566.800 | 29 |  |  |  |

Compared significance under the 0.05 level is indicated by *.

**Table 10.** Post-hoc analysis of the time spent on wayfinding tasks for three different age groups.

| (I) Age Range | (J) Age Range | Average Difference (I–J) | Standard Error | F | Sig. (p) |
|---|---|---|---|---|---|
| 19–30 | 31–44 | −156 | 123 | 1.627 | 0.085 |
| | 45–64 | −227 | 106 | 1.183 | 0.033 * |
| 31–44 | 19–30 | 124 | 115 | 0.763 | 0.082 |
| | 45–64 | −102 | 128 | 1.272 | 0.106 |
| 45–64 | 19–30 | 258 | 109 | 0.852 | 0.021 * |
| | 31–44 | 118 | 132 | 1.045 | 0.117 |

Compared significance under the 0.05 level is indicated by *.

The participants were given a questionnaire after performing the wayfinding tasks, and feedback was given for each task. The options were set as (1) very easy to find—one point; (2) easy to find—two points; (3) ordinary—three points; (4) not easy to find—four points; (5) very difficult to find—five points. The average number of points was calculated based on the responses of the 30 participants, and this was used to assess the degree of difficulty of each wayfinding task. This research analysis mainly selects wayfinding tasks with more than three points as those locations with significant wayfinding problems in the opinions of the 30 participants. The statistical results of wayfinding task difficulty are shown in Table 5. Meanwhile, these results are supplemented by the axial analysis of the space syntax theory to cross-compare the factors that affect the wayfinding behaviors. In the questionnaire feedback, it was found that scores greater than three points were all scattered on the first basement floor of the medical building, including physiological examination room, blood collection station, aisle of urodynamic examination, extracorporeal shock wave lithotripsy room, MRI Room, 64-row computed tomography (CT) room, and angiography room (Table 4). The results show that the route planning of the first basement floor is likely to cause trouble in wayfinding processes for patients visiting the clinic.

The results of the wayfinding experiment show that the first basement floor is the one where most people became lost. The behavioral annotation diagram (Figure 10) shows that the five routes numbered 32, 193, 186, 2, and 8 are the open main routes (Table 2), and they are also the most common routes on which people become lost. Compared with the axial analysis of space syntax theory, its global integration value, Rn, is a moderate value and the number of branch routes connected to each of these five open main routes exceeds six. This result validates Wang's research [19], which pointed out that when the longest moving route has a moderate Rn value (>6), it is likely to cause a higher proportion of lost wayfinding behaviors.

In addition, from the five open main routes in Table 3 (routes 32, 193, 186, 8, and 2) and the behavioral annotation diagram and lost points on the first basement floor of the medical building in Figure 10, it can be found that the ring structure formed by the routes 32, 186, 8, and 2 can be regarded as a "symmetrical ring" space pattern. We found that many participants moved back and forth on this ring structure aisle or stayed at corners for a long time. This phenomenon of becoming lost that was seen with the participants is consistent with Liu's research [33], which pointed out that a "symmetrical ring" spatial pattern is more likely to become a maze. Therefore, it is recommended that when designing and planning the guiding signage system for medical buildings, special attention should be paid to whether there is any confusion caused by the ring structure. The results of the wayfinding tasks and space syntax analysis presented in this study have consistent conclusions with regard to the locations where it is easy to become lost.

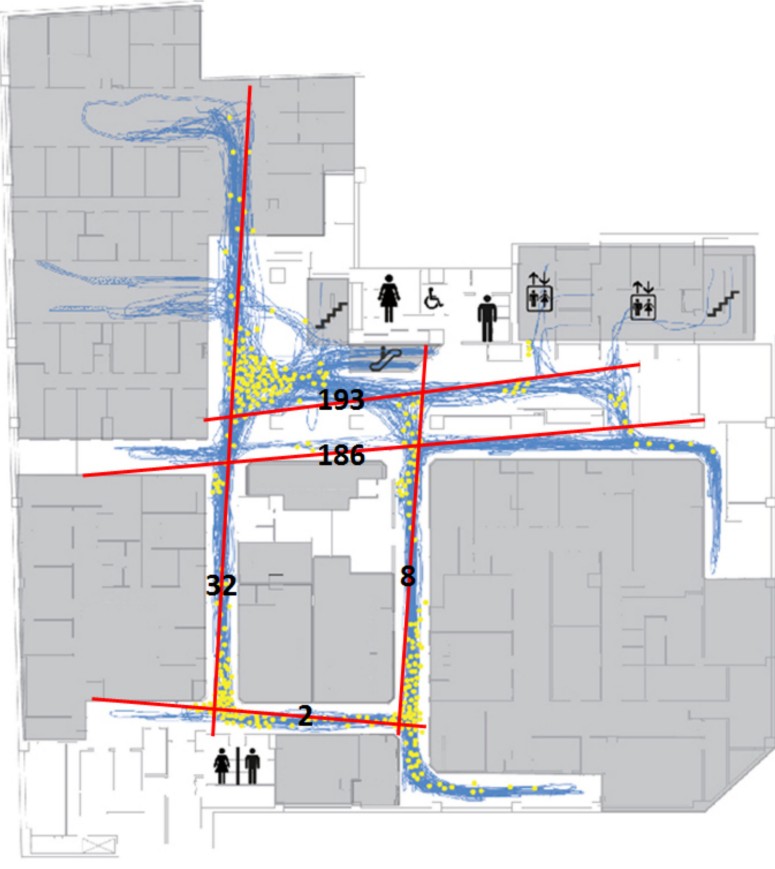

**Figure 10.** The behavioral annotation diagram and points where participants became lost on the first basement floor of the CCH/CKB Building.

### 4.4. Isovist Analysis and Signage Planning

The relationship between space syntax and the signage types is mainly based on the degree of the visibility, as obtained by the results of isovist analysis and the size of the surrounding walls of the location. In principle, if the location has a wide visibility range and there are not too many walls, it is recommended to set up hanging signage. In contrast, if there is a large area of the wall, the wall-mount type of signage is recommended. As for spaces with a narrow field of view, the protruding type is recommended for these. If there are not too many walls at such locations, it is recommended that free stand signage is used. The spatial characteristics should be taken into account with regard to the type of wayfinding system so that the appropriate type and content of the signage are used. In the case of the first basement floor, where the current route planning is more likely to cause confusion and disorientation to the public, this can be easily seen from the major route isovist analysis (Figure 11). The main visual dislocation is in the area surrounded by the walls of the aisle with route 34-32-186, followed by the intersection with route 186-8, then the intersection with route 32-2, and finally the intersection with route 2-8. However, the signage is configured densely at the five routes (routes 32, 193, 186, 8 and 2), and this could easily cause confusion. In line with Macminner [30], it is proposed that there should not be too many signs here to avoid visual confusion caused by too many directories, so the signage should be placed at appropriate distances at decision points with easy-to-read, direct and conspicuous icons and contents, and the overall spatial design of the wayfinding system should be consistent.

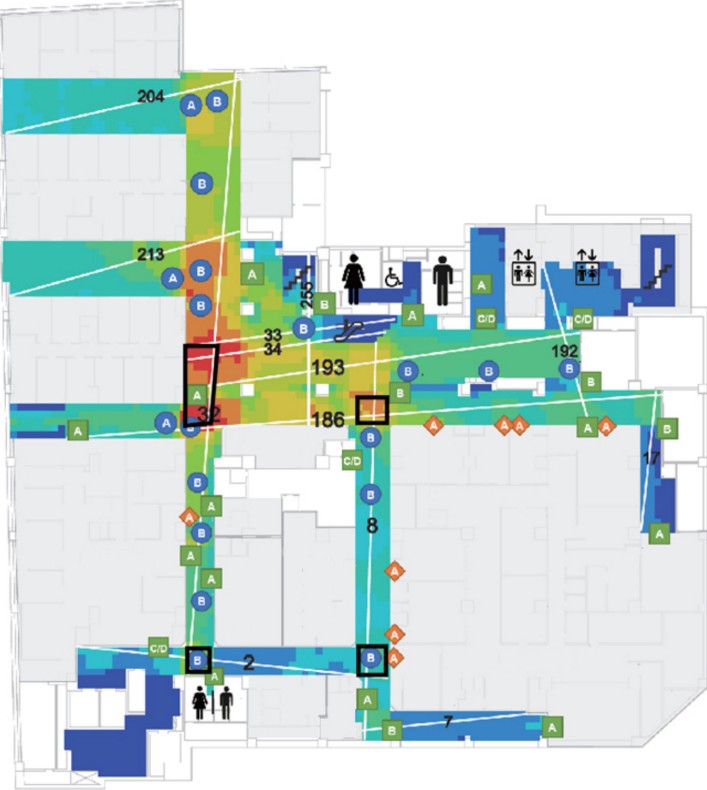

**Figure 11.** The overlapping map of the signage, axis and route isovist on the first basement floor of the CCH/CKB Building.

In Figure 11, the escalator leads to the first basement floor, and the first location is the open space formed by the routes 34, 32, 193 and 255, and the spatial characteristics of this location are transitional areas for selecting the main direction of movement via this position. The Depthmap calculates a 360° field of view at the subject location to obtain the visual field of this area. The main visual dislocation is in the area surrounded by the walls of the clinics and the route 34-32-186. The signage here should be a wall-mounted first basement floor plan, so that the patients can quickly determine the direction of their destination.

In Figure 12, because of the intersection (point A) of the route number 186-8, and the fact that the spatial characteristics of this area are cross-shaped and located in the secondary visual axis staggering area, the sign type here should be an up hanging guiding sign, and the guiding content should be the identification space and examination room that the route mainly belongs to, and then supplemented by the target space and examination room that the secondary route belongs to, to show the primary and secondary structures of different routes. The main and secondary structures of the different routes are presented to facilitate the identification of the destination routes by the patients. The spatial characteristics will lead the patients to turn right intuitively and proceed to the intersection of route number 32-2 (point B) and follow route number 32 in the direction of number 2, which is a T-shaped space. Therefore, the wall at the intersection should be set up with wall-mounted guiding directories, and the guiding contents should be mainly the identification space and examination room that the route will pass through, so as to help people to reach the destination quickly. The up hanging signage along route 32 should be removed, and the location of the inspection room should be marked by protruding identification signage on the door of the inspection room. Finally, the intersection (point C) with the intersection of the route 2–8 is a T-shaped intersection, but there is also a route number 7 further down along route number 8, so the signage type that should be set up at this intersection should be an up hanging sign, and the guidance content should be the identification space and examination room that the route will pass through, and then the target space and

examination room that the secondary route belongs to should be added. The primary and secondary structures of the different routes will be displayed to facilitate the identification of the destination by the patients.

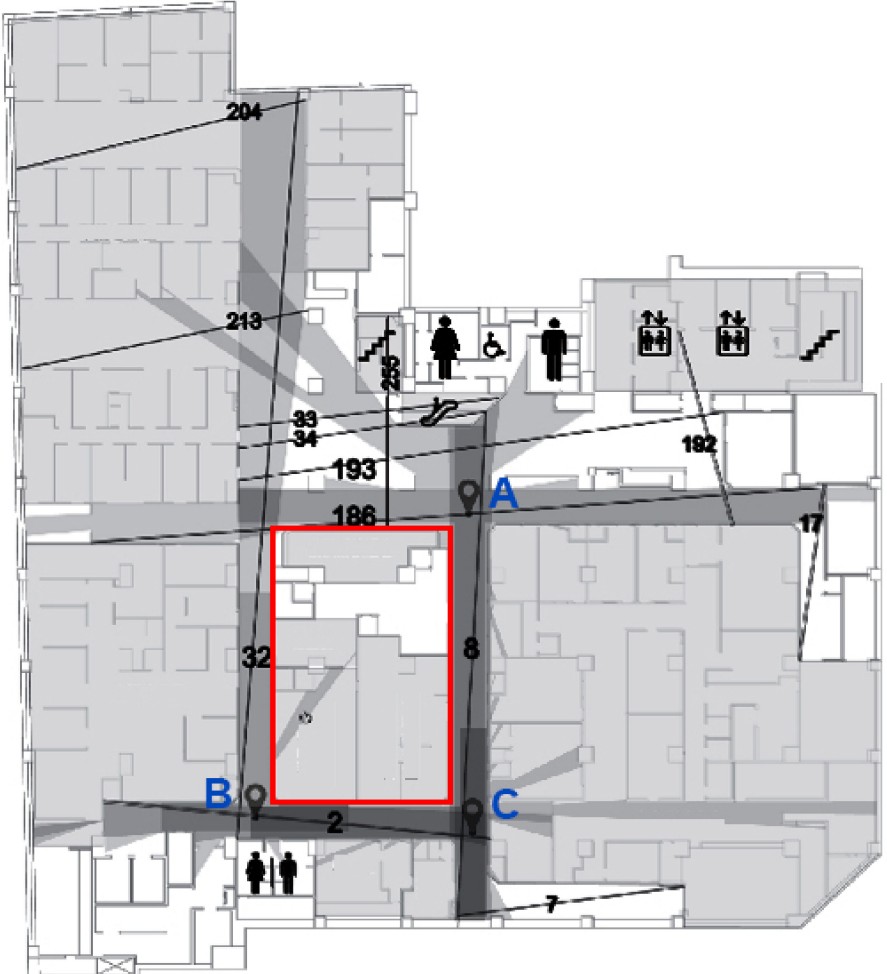

**Figure 12.** The visual area of the subject location in the intersection area of route numbers 186, 32, 2, and 8.

Next, Depthmap software is used for the isovist analysis, and we can see that the escalator's route 34 reaches the first floor of the basement, and the first thing is the open area formed by routes 33, 34, 193 and 255 (Figure 12). When the spatial characteristics of this area are not clear enough, it is easy to be confused and not know how to move. In addition, for the ring-shaped area formed by the routes 32, 2, 186 and 8 the two opposite sides are of equal length, that is, there is a "symmetrical ring" spatial structure (Figure 12, red rectangular area), making the spatial system of this area too homogeneous, and thus, it is difficult for people to identify the various routes and so they can easily become lost. It is recommended that a good outpatient area plan should be a symmetrical tree plan with clear indicators in areas that are easy to become lost in, as this will provide clear and precise information to the public.

## 5. Conclusions

This study investigated the relationships between the spatial configuration of the outpatient areas of health care facilities and the wayfinding behavior of patients. It also examined whether the results of the participants' wayfinding tasks were consistent with the analysis results predicted by the space syntax analysis. In addition, this study applied axial mapping analysis and isovist analysis to find out the locations of visual axis staggering and

suggest the location and type of content of the signage to optimize the functional efficiency of the guiding signage system.

When people are in an unfamiliar environment, they first receive environmental information through their vision. They then form their own knowledge of the environment through internalization of this and complete the process of interpreting the environment they are in. In health care facilities, when patients have problems with finding their way around, they are guided by the guiding signage system provided by the health care facilities to help them solve such problems. Good spatial planning can reduce the wayfinding problems in a clinic. The final results of this study showed that in the wayfinding task experiment gender had no significant impacts on wayfinding efficiency, while there was a significant impact due to age. The older people needed much more time to complete the wayfinding tasks, which means that they had a poorer performance with regard to wayfinding efficiency. In terms of space syntax analysis, the second-floor outpatient areas of the research field have a tree structure with the most user-friendly route plan for the patients, followed by the first-floor outpatient areas, and the least user-friendly were the route plans of the first basement floor outpatient areas, where patients were easily disoriented due to the interlocking route plans forming a symmetrical ring structure. Therefore, in a symmetrical circular aisle, special attention should be given to the visual concentration areas to guide the patients to their destinations by placing appropriate signage forms and contents that match the characteristics of the space. Meanwhile, the text of the sign should not be too small and should not convey too much information. In addition, different colors should be used on the walls to distinguish the operation properties of the space, and an appropriate wayfinding system planning approach should be applied to provide clear and precise guidance information to the patients. A well-planned wayfinding system can facilitate the public's awareness of the structure of the map, and thus reduce problems when going to a destination. However, the results of this study show that the planning of the wayfinding system should be based on the minimal signage configuration required to meet the needs of the patients. Too many signs make it difficult for the patients to immediately determine the nearest route to the destination. In areas where it is easy to become lost the text of the signs should not be too small or convey too much information, as this makes it difficult for people to understand them and causes confusion. As a public space, the overall structure and movement of medical buildings must comply with the relevant safety regulations. Therefore, since the structure of the indoor space cannot be changed at will, the spatial characteristics of the space and movement should be strengthened in areas where people easily become lost, so that the patients can reach their destinations quickly.

In order to objectively analyze the impact of the original architectural structure design and planning of the medical building, and the guiding signage system set up later, on the wayfinding behaviors of the general users of the hospital, the experimental object of this study was members of the general public who had never been to the experimental site (CCH/CKB Hospital) before. With regard to the individuals who will come to a hospital, some patients, such as those with dementia, will face greater problems with regard to wayfinding behaviors, and further analysis and research must be carried out for this group. Therefore, this study's limitations are that it focuses on the general public, and that the experimental field is also limited to the outpatient area (B1 to F2) in the hospital. Future research will thus focus on the wayfinding behaviors of special groups or patients (e.g., dementia patients, etc.) and extend to the ward areas on the other floors.

However, we have achieved the purpose of this paper with our final results, and it is hoped that these will help solve the wayfinding problem in the focal hospital and improve the quality of the hospital environment. The results of this study can also be used as a reference for other health care facilities in the future when planning and constructing related guiding signage systems.

**Author Contributions:** Conceptualization, Ming-Shih Chen, Yao-Tsung Ko and Wen-Che Hsieh; data curation, Yao-Tsung Ko and Wen-Che Hsieh; resources, Ming-Shih Chen and Yao-Tsung Ko; funding ac-quisition, Ming-Shih Chen; investigation, Yao-Tsung Ko and Wen-Che Hsieh; methodology, Yao-Tsung Ko and Wen-Che Hsieh; software: Yao-Tsung Ko and Wen-Che Hsieh; visualization, Yao-Tsung Ko and Wen-Che Hsieh; project administration, Ming-Shih Chen; writing—original draft, Ming-Shih Chen, Yao-Tsung Ko and Wen-Che Hsieh; writing—review and editing, Ming-Shih Chen and Yao-Tsung Ko. All authors have read and agreed to the published version of the manuscript.

**Funding:** This research was funded by the Ministry of Science and Technology of Taiwan under grant MOST 105-2410-H-029-035.

**Institutional Review Board Statement:** Not applicable.

**Informed Consent Statement:** Not applicable.

**Data Availability Statement:** The data presented in this study are available on request from the corresponding author.

**Conflicts of Interest:** The authors declare no conflict of interest.

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
