# Peer review of "Exploring the Planning and Configuration of the Hospital Wayfinding System by Space Syntax: A Case Study of Cheng Ching Hospital, Chung Kang Branch in Taiwan"

_ijgi, doi:10.3390/ijgi10080570_

Round 1

Reviewer 1 Report

This is a fantastic paper on a complex topic; its clear in it's structure and message; the included figures are a very helpful to link text and graph; the most surprising results was that the capacity of wayfinding was significantly different with regard to age but not biological sex (gender in the article). 

Unfortunately, what is really missing (and was at least expected in the conclusion) is a reflection of the difference between 'experimental participants' and 'real patients'. Having worked at different University Hospitals for a long time, I could say 'I would know from experience'. There is a difference in getting around when for example you're being anxious b/c a family member/a friend just had an emergency and you run into the clinic, to find the person. Furthermore, often enough people that are hospitalised are cognitive impaired (I myself do research with people with symptoms of dementia, and we find they are usually ignored in way finding studies).

It therefore, would be of interest how 'real patients' (either getting there for a planned meeting/treatment etc. or trying to get around in an emergency situation) experience the suggested 'tree structure' and 'symmetrical circular aisle'. 

It would be great if the conclusion could say something about the 'real patients' , what to expect/to needs to be done/existing research gaps' etc. 

Reviewer 2 Report

The study mainly describes the steps in the analysis of the signage located in a hospital with the aim of finding weaknesses in this wayfinding system and improvement suggestions. The analysis is based in space syntax. A second part of the study tests experimentally how individuals oriented themselves in the facilities of the hospital that were analysed and compared both types of information, that is, the theoretical and experimental data.

The experimental approach adds value to the study.

Limitations are: the aims are confusing, the procedures of the experimental part and the statistics need improvement, and some objectives do not match what was actually tested.

Mayor issues:

1) The state of the art is scarce in relation to the second part of the study. The paper reports wayfinding behavior and included aims in relation to this issue but little background is reported about the wayfinding behavior. I think it would be important to contextualize previous results on the field because the authors aimed to test the effect of both gender and age.

2) Partially Related with point 1): the main aim of the behavioral study is to compare the results with the predictions made in the first part of the study. The reason for making gender and age comparisons is not substantiated. In addition, the division into age groups with the small number of participants does not make sense. Do gender and age comparisons really contribute to this study? Furthermore, there are many previous studies that answer this question.

3) The statistical tests used make sense in parametric distributions, however, the number of participants could indicate a non-parametric distribution. This issue is not considered. The following must be done: to check that the sample follows a normal distribution and homogeneity of variances. There are tests to check it. If data set can assume both a normal distribution and homogeneity of variances, the tests already carried out could be maintained, if not, non-parametric analyzes should be performed.

4) How did the authors estimated the required sample size?

Minor issues:

1) Line 111: please, explain "Rn"

2) Line 229: "we firstly analyzed and annotated the signage configuration.." please explain how this process was done

3) Tables must have sufficient detail to be interpreted regardless of the main text. Table 1: please, explain Rn and CN. Table 4: What do authors refer to in the table by "difficulty points for wayfinding tasks"? What is specifically measured and where is obtained from (questionnaire feedback)?

4) The authors should more specifically determine the sample characteristics and the recruitment of the individuals.

5) The authors should more specifically determine the instructions given to the participants and how it was recorded ("observed and record", line 341). What was recorded (e.g. written notes, a video recording)? What was quantitatively recorded? How many people acted as judges? Did they have any training?

6) Line 356: The authors cannot claim that women are more efficient, there are no significant differences.

7) Line 378: please, the value of significance should be p=0.028, it makes no sense to add the significance threshold.

8) Line 384-386: I suggest modifying the statement and explaining the possible reasons for longer times in older individuals.

9) Line 421: "disorientation" I suggest modifying the statement and explaining how disorientation was determine in the study.

10) Line 495: please, deleted "correlation", since no correlation is calculated, only relationships are discussed.

11) Study limitations should be discussed.

Reviewer 3 Report

Concepts need to be defined and explained where they first appear. None of these concepts were defined or explained *anywhere* in the paper: Rn, CN, axial mapping analysis, isovist analysis, axial map, tree structure and trunk, global integration value. Space syntax theory needs considerably more and clearer explanation. You cannot assume readers know anything about it. I wondered how you were making recommendations about different signage forms based on different space syntax because you never explained how space syntax relates to signage needs.

Consider the organization of the paper. Your “Conclusion” should really be your introduction since that is where you explain the research purpose and importance. Don’t make the reader wait until the end of the paper to tell them why they should care about your work. Don't talk about correlation when you didn’t run any correlation statistics. You should also move section 3.2 before section 3.1 since you mention the building in 3.1 without explaining it until the following section. And do not repeat whole passages from Section 3.2 in Section 4.2.

You need a lot more detail about your methodology. How were participants recruited? Was there an experimental vs control group? What specific tasks were they assigned? Did they all do the same tasks? Or some of the tasks? Were they assigned at random? Did they all do the same tasks in the same order or a random order? What precise procedures were used to conduct the experiment? Were the cameras video or still, and how was that data reviewed and analyzed? Other details that were missing:

  • How was “convenience” of each route determined? This needs to be explained in *detail* since it is a crucial part of the study. 
  • Explain how the route numbers were assigned.
  • How was the difficulty points for wayfinding tasks calculated (Table 4)?

In Section 4.3, you might consider adding a table with the total signs by category and % of all signs. That would help the reader interpret the maps in Figure 5.

Tables 1, 2, and 3 are pretty confusing since there is no explanation of what order the lines are listed in. They don’t seem to be in order by rank or Rn. Either of those two columns would be useful ways to organize the data in the tables.

A few arguments were totally unjustified:

  1. On what are you basing the statement that “There are relatively few studies adopted computer analysis” (line 143)? There is no evidence to support that statement in your paper. How many databases did you search? What search terms did you use? If you want to make this argument, you need to justify it. 
  2. You also need to be careful in your findings not to overstate what you can prove based on n=30. Thirty people is not enough to say that women are more efficient than men. 
  3. I also wondered about the argument regarding older people being slower wayfinders. Did you consider that they may actually walk more slowly? Did you time people’s walking speeds? Because then you could control for that variable. But if you didn’t, perhaps you shouldn’t just assume older people are slower at wayfinding when they might just be walking slowly because they are older.
  4. In the conclusion, you start talking about psychological burden. There was nothing in your experiment that tested psychological burden so you really have no basis for the statement.

There were a number of poor word choices

  • “a perfect framework” (line 49) - Perfection is never the goal.
  • “wayfinding situations” (line 59) - Do you mean wayfinding problems? Challenges? Situations is really not specific enough.
  • “resulted in half-hearted efforts” (line 145) - Unless you are intentionally trying to alienate and anger other researchers in wayfinding, you need to reword this in a much less offensive manner.
  • Section 4.1, the text says “identification” but Figure 3 says “recognition"; the text says “orientation” but Figure 3 says “directions.” These all need to match correctly.
  • “the smaller the number of numbers listed” (line 269) - This makes no sense at all. Please clarify.

Round 2

Reviewer 2 Report

I appreciate that the author has improved this work. I have nothing to add to my previous comments.